# On-Road Object Importance Estimation: A New Dataset and A Model with Multi-Fold Top-Down Guidance

Zhixiong Nan[1], Yilong Chen[1], Tianfei Zhou [*2], and Tao Xiang[1]

[1]College of Computer Science, Chongqing University, Chongqing, China.
[2]School of Computer Science and Technology, Beijing Institute of Technology, Beijing, China.
nanzx@cqu.edu.cn, chenyilong@stu.cqu.edu.cn, tfzhou@bit.edu.cn,
txiang@cqu.edu.cn [†]

## Abstract

This paper addresses the problem of on-road object importance estimation, which utilizes video sequences captured from the driver's perspective as the input. Although this problem is significant for safer and smarter driving systems, the exploration of this problem remains limited. On one hand, publicly-available large-scale datasets are scarce in the community. To address this dilemma, this paper contributes a new large-scale dataset named Traffic Object Importance (TOI). On the other hand, existing methods often only consider either bottom-up feature or single-fold guidance, leading to limitations in handling highly dynamic and diverse traffic scenarios. Different from existing methods, this paper proposes a model that integrates multi-fold top-down guidance with the bottom-up feature. Specifically, three kinds of top-down guidance factors (*i.e.*, driver intention, semantic context, and traffic rule) are integrated into our model. These factors are important for object importance estimation, but none of the existing methods simultaneously consider them. To our knowledge, this paper proposes the first on-road object importance estimation model that fuses multi-fold top-down guidance factors with bottom-up feature. Extensive experiments demonstrate that our model outperforms state-of-the-art methods by large margins, achieving **23.1**% Average Precision (AP) improvement compared with the recently proposed model (*i.e.*, Goal).

## 1 Introduction

According to the World Health Statistics of WHO (34), road traffic injuries account for a significant 29% of all injury deaths, with nearly 1.3 million people losing their lives in traffic accidents annually. Accurately estimating the importance of on-road objects can pave the way for safer (*e.g.*, automatic emergency braking (45; 39)) and smarter driving systems (32; 25; 38; 49; 4; 36; 11), potentially preventing numerous accidents.

Although on-road object importance estimation presents significant research value, it has not been widely explored. One main reason is the scarcity of publicly-available large-scale datasets in the community. Specific for the on-road object importance estimation task, the only one publicly-available dataset is Ohn-Bar *et al.* (33), which contains 3,187 frames, 8 scenes, and 16,076 object importance annotations. Such small-scale dataset supports to train small and less complex models.

---

[*]Corresponding author.

[†]This work is supported by Chongqing Natural Science Foundation Innovation and Development Joint Fund (CSTB2023NSCQ-LZX0109).

However, traffic scenes are dynamic and diverse, asking for complex models to handle various traffic situations. Some researchers have recognized this dilemma and propose some datasets (8; 21; 46). Unfortunately, these dataset are not publicly-available, thereby the dilemma has not been fundamentally addressed. To fundamentally address this dilemma and push forward the advancement of on-road object importance study, this paper will release a large-scale dataset (named as **TOI**, Traffic Object Importance) containing 9,858 frames, 28 scenes, and 44,120 object importance annotations. Compared to Ohn-Bar (33), **TOI** achieves a 3.1-fold increase in frames count, a 3.5-fold increase in scene count, and a 2.7-fold increase in object count.

From the perspective of methodology, some methods have been proposed (21; 8; 50; 33). However, these methods are relatively simple, exhibiting the low performance when confronting to challenging traffic scenarios. This motivates us to think about a question: why can not existing methods perform well? The potential reason is that existing on-road importance estimation methods underestimate the complexity of traffic scenarios, individual bottom-up mechanism (50) (assuming important objects are the objects with salient color, texture, size, *etc.*) or simple top-down guidance mechanism (8; 21) (fusing the bottom-up information with a certain type of top-down information such as semantics, ego-car trajectory, driving task (27), *etc.*) can hardly address dynamic and diverse scenarios.

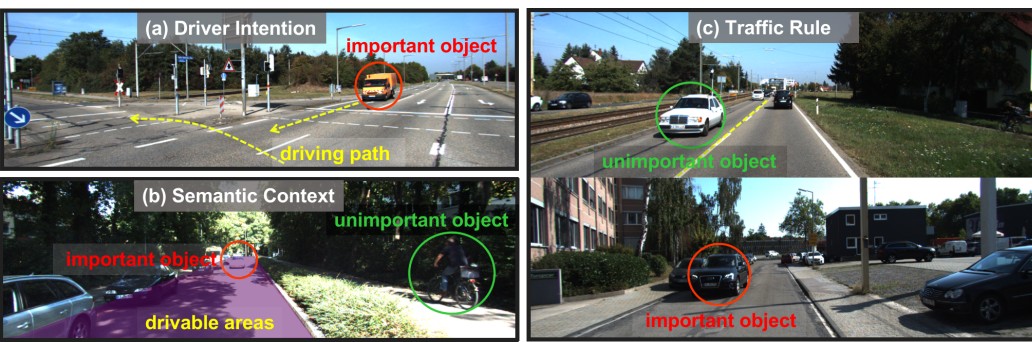

Figure 1: The crucial factors considered by human drivers when estimating on-road object importance.

Therefore, a smarter model is needed. Inspired by the fact that a human driver can accurately estimate object importance in challenging situations, this paper attempts to design a model by analysing the human reasoning mechanism during estimating object importance. To this end, the primary question is "*what essentially crucial factors are considered when a human driver is estimating the importance of objects?*". **Firstly**, *the attributes (e.g., size, color, and texture) of the object* is considered. For example, when a truck with the big size and a car with the small size simultaneously appear in front of the ego-car, the truck is more important since it imposes bigger impact on the driving. **Secondly**, *the driver intention* is considered. The objects that will riskly collide with the ego-car intention driving path or the objects locating on the ego-car intention driving path present high importance, as shown in Fig. 1a. **Thirdly**, *overall semantic context of the whole traffic scene* is considered. A human usually pay more attention on the objects in drivable areas rather than the objects in undrivable areas. As shown in Fig. 1b, the person riding a bicycle in undrivable areas is unimportant. **Fourthly**, *traffic rule* is considered. Most of traffic participants obey the traffic rule, thus the traffic rule is an critical factor for a human to estimate object importance. For example, as shown in Fig. 1c, when there exists a lane marking between the oncoming car and the ego-car, the human driver may consider the oncoming car as unimportant. In contrast, if there is no lane marking between them, the importance of the oncoming car significantly increases. The traffic rule is crucial for object importance estimation. However, none of existing methods utilizes traffic rule to estimate on-road object importance.

Based on the above observations, we propose a model with multi-fold top-down guidance including *driver intention*, *semantic context*, and *traffic rule*. As far as we know, it is the first on-road object importance estimation model that fuses multi-fold top-down guidance factors with the bottom-up feature. The proposed model consists of two kinds of pathways (*i.e.*, bottom-up pathway and top-down pathway). Bottom-up pathway and top-down pathway are fused to estimate object importance. Specifically, in the top-down pathway, the top-down guidance factors of *driver intention* and *semantic context* are involved in the proposed Driver Intention and Semantics Guidance (**DISG**) module, and *traffic rule* is modeled in the proposed Traffic Rule Guidance (**TRG**) module. In the bottom-up pathway, Object Feature Extraction (**OFE**) module is proposed to extract object features in both spatial and temporal dimensions.

A series of comparison and ablation studies are conducted on a public dataset (33) and our **TOI** dataset. The comparison experiment results show that our model has a solid advantage over the baselines. The ablation study results validate the effectiveness of our proposed interactive bottom-up & top-down fusion framework and multi-fold top-down guidance modules (*i.e.*, **DISG** and **TRG**).

The contributions of this paper are as follows. **1)** This paper contributes a new large-scale dataset, which will be publicly released. This dataset is almost three times larger than the current publicly available public dataset (33). **2)** This paper contributes an object importance estimation model. As far as we know, it is the first on-road object importance estimation model that integrates multi-fold top-down guidance factors with the bottom-up feature. **3)** The traffic rule is crucial for object importance estimation. However, none of existing methods utilizes traffic rule to estimate on-road object importance. This paper considers the effect of traffic rule on object importance and successfully models this abstract concept by proposing an adaptive object-lane interaction mechanism.

## 2 Related Works

***On-Road Object Importance Estimation Related Datasets.*** Currently, the primary dilemma of research on the on-road object importance estimation is the lack of sufficient data. Among existing datasets relevant to autonomous driving perception tasks, only a few meet the data requirement of providing images from the driver's first-person perspective while also including object importance level labels. Ohn-Bar *et al.* (33) are the first to define the problem of on-road object importance estimation, and they propose a small-scale publicly available dataset, which contains 8 scenes. Gao *et al.* (8) and Li *et al.* (21) have significantly increased the number of scenes. However, their datasets are not publicly available, thus the contributions to the community is limited. Datasets (19; 41; 48) are with detailed annotations such as ego's reaction and road topology. They have great potential to advance the development of on-road object importance estimation. However, currently, they lack object importance level labels and cannot be directly applied to this task.

***On-Road Object Importance Estimation Related Methods.*** Currently, on-road object importance estimation methods can be divided into two categories: 1) methods solely utilizing bottom-up feature; 2) methods utilizing single-fold top-down guidance.

The methods solely utilizing bottom-up feature focus on the visual attributes of the objects. The bottom-up processing method is initially introduced in (44), and Itti *et al.* (17) propose one of the first bottom-up mechanism based models. Following this, many researchers are inspired by this concept (43; 15; 18; 35). Zhang *et al.* (50) introduce a model, which solely relies on RGB clips for object importance estimation. This model employs graph convolutions to characterize the interactions among on-road objects. Nitta *et al.* (30) develop a model that extracts temporal features from optical flow images to infer the states of moving objects. The optical flow images are also used in Malla *et al.* (26) to assess the states of moving objects. The bottom-up methods can also be found in the works (33; 52; 47; 23; 14; 24). Although the bottom-up feature is crucial for importance estimation, the methods solely rely on bottom-up feature can not function well in the complex scenarios.

The importance of an object is influenced by many factors such as driver intention, which cannot be fully utilized through bottom-up methods. However, current methods are relatively simple and relies on single-fold guidance. Niu *et al.* (31) utilize a Transformer with shared weights to identify high-risk objects and generated semantic warning sentences. Li *et al.* (21) investigate the impact of driver intention, employing the action and trajectory data of the ego-car as additional supervisory signals in auxiliary tasks to enhance model performance. Gao *et al.* (8) utilize the driver's goal to estimate object importance. A cause-effect problem was formulated in (20), which introduced a model to estimate the risky object by considering its potential impact on the driver's behavior. Tang *et al.* (42) provide a more comprehensive understanding of how driver intentions under different tasks affect the driver attention. Single-fold top-down guidance can also be found in the works of attention prediction task (7; 16; 22; 6; 1; 29; 5; 28). However, none of these methods utilizes multi-fold top-down guidance factors to estimate the on-road object importance.

## 3 A New Dataset: TOI

We thoroughly review existing datasets for on-road object importance estimation as well as the datasets for the related tasks, and provide a summary in Tab. 1. The datasets for the related tasks (*e.g.*, risk assessment (37; 48; 19), accident anticipation (41), and situation awareness (40)) do not include object importance labels, making them unsuitable for object importance estimation task. Most datasets (*e.g.*, (21; 8)) for object importance estimation are not publicly available. The only

Table 1: Comparison between the TOI and State-of-the-art Datasets. 'Impo.' represents the object importance annotation.

| Dataset | Task | Public | Impo. | Extra-Information GPS/IMU | Lidar | 3D-Labels | Objects | Frames | Scenes | FPS | Year |
|---|---|---|---|---|---|---|---|---|---|---|---|
| HDD (37) | risk assessment | ✓ | ✗ | ✓ | ✓ | ✗ | - | - | - | 30 | 2018 |
| 1361-honda (48) | risk assessment | ✓ | ✗ | ✗ | ✗ | ✗ | - | - | 1,361 | - | 2020 |
| RiskBench (19) | risk assessment | ✓ | ✗ | ✗ | ✗ | ✗ | - | - | 6,916 | - | 2024 |
| NIDB (41) | accident anticipation | ✓ | ✗ | ✗ | ✗ | ✗ | - | 499,500 | 4,995 | - | 2018 |
| A-SASS (46) | situation awareness | ✗ | ✓ | ✗ | ✗ | ✗ | - | - | 10 | 30 | 2022 |
| ROAD (40) | situation awareness | ✓ | ✗ | ✓ | ✓ | ✓ | 560,000 | 122,000 | 22 | 12 | 2023 |
| Ohn-Bar (33) | on-road object importance estimation | ✓ | ✓ | ✓ | ✓ | ✓ | 16,076 | 3,187 | 8 | 10 | 2017 |
| Goal (8) | on-road object importance estimation | ✗ | ✓ | ✓ | ✗ | ✗ | - | 244,980 | 743 | 30 | 2019 |
| Li (21) | on-road object importance estimation | ✗ | ✓ | ✓ | ✓ | ✓ | - | - | - | 2 | 2022 |
| TOI | on-road object importance estimation | ✓ | ✓ | ✓ | ✓ | ✓ | 44,120 | 9,858 | 28 | 10 | 2024 |

publicly available dataset is Ohn-Bar (33), but it is a small scale dataset. In response to the scarcity of publicly-available large-scale datasets for on-road object importance estimation, we contribute a large-scale dataset named **TOI** (Traffic Object Importance). **TOI** is built by re-annotating the authoritative KITTI (9) dataset. While there are many datasets (*e.g.*, nuScenes (2; 40; 37)) that be used for object importance annotations, we select KITTI dataset for the following reason: KITTI is the worldwide benchmark in the field of autonomous driving. In addition, KITTI is collected in diverse real traffic scenes including rural areas and on highways with rich date formats, making the dateset friendly for various tasks.

***Annotation Procedure.*** The criterion of object importance might be ambiguous as different drivers usually hold different opinions on the importance judgment. Currently, object-level importance labels are annotated without checking mechanism. Although multiple annotators perform the annotations, the annotations finished by the certain annotator are not checked by others, leading to the unreliable and ambiguous annotations. To generate reliable annotations, we adopt two mechanisms: the *double-checking annotation mechanism* and the *triple-discussing annotation mechanism*.

The *double-checking annotation mechanism* operates as follows. Initially, the first annotator (an experienced driver) labels the object importance at every 10 frames. To guarantee the reliability of annotations, the first annotator only selects one object as the important object at each time of observing the whole 10 frames. The annotation for these 10 frames is finished until all important objects are annotated, then the annotator moves to the next set of 10 frames for annotation. Subsequently, the annotation results are checked by the second annotator (who is also an experienced driver). When the second annotator finds a disputed annotation, the first and second annotators discuss together to reach an agreement. If they are unable to reach an agreement, the *triple-discussion annotation mechanism* is activated. In this case, the third annotator is invited to discuss the final annotations.

***Statistics and Comparison.*** Totally, 9,858 image frames are annotated, generating 44,120 objects with the importance or unimportance annotations, among which 5,052 objects are annotated as important. The annotated data are randomly split into training/testing datasets with a ratio of 8,121 : 1,737. The comparison between **TOI** and existing on-road object importance estimation datasets and similar task datasets are presented in Tab. 1. Compared to the publicly available Ohn-Bar (33) dataset, **TOI** represents the significant advantages in following three aspects. **Frame quantity**: **TOI** exhibits a substantial increase in the number of frames, with 9,858 frames compared to 3,187 frames in the Ohn-Bar dataset. **Object quantity**: the number of annotated objects is 44,120 compared to 16,076 in the Ohn-Bar dataset. **Scene diversity**: while Ohn-Bar contains only 8 scenes, **TOI** covers 28 scenes. Compared to Goal (8) dataset, **TOI** has rich annotations such as Lidar and 3D tracklet labels. The abundance of multimodal annotations enables **TOI** to support the research on on-road object importance estimation using multimodal learning methods in the future. Though Goal (8) presents the advantage in terms of frame number and scene diversity, it is not publicly available. Compared to Li dataset (21), **TOI** offers the frame rate of 10 FPS. This high temporal resolution is critical for capturing the dynamic changes of on-road object importance.

***Annotation Challenges.*** Compared to datasets for other tasks, **TOI** may not be considered large-scale, it is relatively large-scale compared to existing publicly available datasets for on-road object importance estimation. However, annotating object importance at this scale is challenging. Each annotation requires multiple annotators and undergoes rigorous checking to achieve satisfactory results. In addition, to generate reliable annotations, only one object is annotated at each time observing the whole video sequence, a complete re-observation of the whole video sequence is required for the annotation of the next object. Moreover, object importance is affected by multiple factors, which imposes difficulties on the annotation.

# 4 Approach

## 4.1 Overview

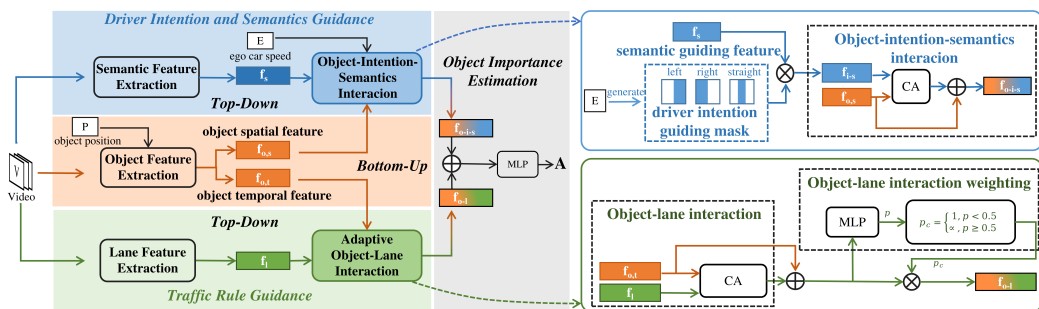

Figure 2: The overview of multi-fold top-down guidance aware model.

Consider a traffic scenario with $N$ on-road objects in $T$ time steps, the goal of this work is to estimate on-road object importance ($A$) at the final time step (*i.e.*, $t = T$) using the video sequence ($V$) captured from the driving perspective over $T$ time step and ego-car velocity information ($E$) at the first time step (*i.e.*, $t = 1$), which is formulated as:

$$A = \mathcal{N}(V, E), \tag{1}$$

where $\mathcal{N}$ represents an on-road object importance estimation network, $V = \{V_t\}_{t=1}^{T}$, and $A = \{A_i\}_{i=1}^{N}$.

In order to effectively fuse multi-fold top-down guidance factors (*i.e.*, *semantic context*, *driver intention*, and *traffic rule*) with bottom-up object visual feature, we propose a multi-fold top-down guidance aware model, the overview of which is illustrated in Fig. 2. Our model is composed of four key modules: **Object Feature Extraction** (**OFE**) module detailed in § 4.2, **Driver Intention and Semantics Guidance** (**DISG**) module described in § 4.3, **Traffic Rule Guidance** (**TRG**) module explained in § 4.4, and **Object Importance Estimation** module introduced in § 4.5.

Firstly, **OFE** extracts object spatial feature $f_{o,s}$ and object temporal feature $f_{o,t}$ from $V$. Then, **DISG** takes $E$, $V$, and $f_{o,s}$ as inputs, and outputs object-intention-semantics interaction feature $f_{o\text{-}i\text{-}s}$. Meanwhile, in **TRG**, the lane feature $f_l$ and $f_{o,t}$ are processed by **adaptive object-lane interaction** mechanism to produce the object-lane interaction feature $f_{o\text{-}l}$. Finally, $f_{o\text{-}i\text{-}s}$ and $f_{o\text{-}l}$ are used to estimate object importance $A$.

## 4.2 Object Feature Extraction

The goal of **Object Feature Extraction** (**OFE**) module is to extract object features in both temporal and spatial dimensions. The input of **OFE** is a RGB video sequence $V \in \mathbb{R}^{T \times 3 \times W \times H}$, and the outputs are object temporal feature $f_{o,t}$ and object spatial feature $f_{o,s}$.

To begin with, **OFE** takes $V$ and $M$ ($M$ denote optical flow images derived from $V$) as inputs to extract the object visual feature $f_v \in \mathbb{R}^{N \times T \times C \times W' \times H'}$ (reflecting the appearance of the object) and the object motion feature $f_m \in \mathbb{R}^{N \times T \times C \times W' \times H'}$ (reflecting the movement of the object). This procedure is formulated as:

$$f_v = \text{Roi}(\mathcal{N}_V(V)), \quad f_m = \text{Roi}(\mathcal{N}_M(M)), \tag{2}$$

where $\mathcal{N}_V$ and $\mathcal{N}_M$ represent the two ResNet18 (12), Roi denotes the ROI pooling (10), $C$ represents the number of channels, $W'$ and $H'$ denote the width and height obtained through ROI pooling.

Subsequently, object spatial feature $f_{o,s} \in \mathbb{R}^{N \times 2C \times W' \times H'}$ is obtained based on $f_v$ and $f_m$. The goal of $f_{o,s}$ is to focus on the spatial information of objects. Therefore, an average pooling is applied on the time dimension (*i.e.*, the dimension of $T$) of $f_v$ and $f_m$, and a self-attention mechanism is utilized to emphasize the spatial information (*i.e.*, the dimensions of $W'$ and $H'$), which is denoted as follows:

$$f_{o,s} = \mathcal{N}_{mhsa}(\text{Concat}(\text{Avg}(f_v), \text{Avg}(f_m))), \tag{3}$$

where Avg denotes average pooling, Concat is concatenation. $\mathcal{N}_{mhsa}$ represents the multi-head self-attention mechanism, and it has the same meaning in the following parts.

Meanwhile, object temporal feature $\boldsymbol{f}_{o,t} \in \mathbb{R}^{N \times C'}$ is also extracted based on $\boldsymbol{f}_v$ and $\boldsymbol{f}_m$. The goal of $\boldsymbol{f}_{o,t}$ is to focus on the temporal feature of objects. Therefore, the dimensions of $\boldsymbol{f}_v$ and $\boldsymbol{f}_m$ are firstly reshaped from $N \times T \times C \times W' \times H'$ to $N \times T \times (C \times W' \times H')$, and then two LSTM networks are applied to extract the temporal feature. Finally, to subsequently fuse $\boldsymbol{f}_{o,t}$ with $\boldsymbol{f}_l \in \mathbb{R}^{N \times C'}$, it is necessary to transform the channel number of $\boldsymbol{f}_{o,t}$, hence a Linear layer is required. This procedure is formulated as:

$$\boldsymbol{f}_{o,t} = \text{Linear}(\mathcal{N}_{mhsa}(\text{Concat}(\mathcal{N}_{lstm}(\boldsymbol{f}_v), \mathcal{N}_{lstm}(\boldsymbol{f}_m)))), \tag{4}$$

where $\mathcal{N}_{lstm}$ is the LSTM network (13), which outputs features for $T$ time steps, and $\boldsymbol{f}_{o,t}$ is computed by indexing the information at the $T$-th time step.

### 4.3 Driver Intention and Semantics Guidance

The driver intention and semantic context significantly affect the on-road object importance in a top-down manner, thus we propose the **Driver Intention and Semantics Guidance** (**DISG**) module. **DISG** consists of two components, namely **semantic feature extraction** and **object-intention-semantics interaction**. The former extracts the *semantic guiding feature* (*i.e.*, $\boldsymbol{f}_s$ in Eq. (5)) from driving scenarios, and the latter uses $\boldsymbol{f}_s$ and *intention guiding mask* (*i.e.*, $\boldsymbol{m}$ in Eq. (7)) to guide the refinement of $\boldsymbol{f}_{o,s}$. In the following part, we will sequentially introduce the calculation processes for *semantic guiding feature* and *intention guiding mask*.

*Semantic guiding feature.* The goal of $\boldsymbol{f}_s \in \mathbb{R}^{1 \times 2C \times W' \times H'}$ is to guide the model to be aware of the semantic context of on-road objects, which is extracted by:

$$\boldsymbol{f}_s = \mathcal{N}_G(\boldsymbol{G}), \tag{5}$$

where $\mathcal{N}_G$ denotes a ResNet18 backbone network and $\boldsymbol{G}$ represents semantic segmentation maps obtained from $\boldsymbol{V}_T$.

*Intention guiding mask.* The goal of $\boldsymbol{m} \in \mathbb{R}^{W' \times H'}$ is making the model be aware of the region that is consistent with the driver intention. However, it is difficult to realize since the driver intention is an abstract concept and the limited information regarding the driver intention is known. Additionally, it is not reasonable to assume the driver intention is known in advance. Therefore, we use three common intention behaviors in driving to reflect the driver intention (*i.e.*, *turning left*, *going straight*, and *turning right*). Intention behaviors are formulated as the corresponding intention guiding masks:

$$\boldsymbol{m}_l = \begin{bmatrix} a & \dots & a & b & \dots & b \\ a & \dots & a & b & \dots & b \\ \vdots & \vdots & \vdots & \vdots & \vdots & \vdots \\ a & \dots & a & b & \dots & b \end{bmatrix}_{(W' \times H')}, \boldsymbol{m}_s = \begin{bmatrix} a & \dots & a & b & \dots & b & a & \dots & a \\ a & \dots & a & b & \dots & b & a & \dots & a \\ \vdots & \vdots & \vdots & \vdots & \vdots & \vdots & \vdots & \vdots & \vdots \\ a & \dots & a & b & \dots & b & a & \dots & a \end{bmatrix}_{(W' \times H')}, \boldsymbol{m}_r = \begin{bmatrix} b & \dots & b & a & \dots & a \\ b & \dots & b & a & \dots & a \\ \vdots & \vdots & \vdots & \vdots & \vdots & \vdots \\ b & \dots & b & a & \dots & a \end{bmatrix}_{(W' \times H')} \tag{6}$$

where $\boldsymbol{m}_l$, $\boldsymbol{m}_s$, and $\boldsymbol{m}_r$ are manually engineered masks to emphasize the information in the right, center, and left regions of the images in the video sequence, respectively. Their sizes are aligned with the size of $\boldsymbol{f}_s$. $a$ and $b$ represent pre-determined low and high values, respectively. We note that $\boldsymbol{m}_l$ representing turning left behavior is allocated with higher value at the right part, which is inspired by the finding of Tang *et al.* (42) demonstrating that when a car is turning left, the driver pays more attention to the right side. Before proposing this predefined intention guiding mask, we design a learnable mask with random initialization to make the model automatically learn the mask to reflect intention behaviors. However, this strategy does not work. The potential reason is that the intention is abstract to learn.

To automatically select the $\boldsymbol{m}$ of corresponding driving behavior, we design the following logic based on the angular velocity $E$ of ego-car:

$$\boldsymbol{m} = \begin{cases} \boldsymbol{m}_l, & \text{if } E > \beta \\ \boldsymbol{m}_r, & \text{if } E < -\beta \, , \\ \boldsymbol{m}_s, & \text{otherwise} \end{cases} \tag{7}$$

where $\beta$ represents the driver turning threshold.

After obtaining $\boldsymbol{f}_s$ and $\boldsymbol{m}$, **DISG** module fuses them and uses the fused feature to guide the refinement of $\boldsymbol{f}_{o,s}$. This procedure is implemented by the **object-intention-semantics interaction** component. The first task of **object-intention-semantics interaction** is to fuse $\boldsymbol{f}_s$ and $\boldsymbol{m}$, and generate the intention-semantics interaction feature $\boldsymbol{f}_{i\text{-}s} \in \mathbb{R}^{1 \times 2C \times W \times H}$, which is denoted as follow:

$$\boldsymbol{f}_{i\text{-}s} = \boldsymbol{f}_s \times \boldsymbol{m}. \tag{8}$$

where the operator $\times$ makes the model pay more attention to the semantic context in the driver intention regions.

The second task of **object-intention-semantics interaction** is to refine $\boldsymbol{f}_{o,s}$ by interacting with $\boldsymbol{f}_{i\text{-}s}$, which is formulated as follow:

$$\boldsymbol{f}_{o\text{-}i\text{-}s} = \mathcal{N}_{mhca}(\boldsymbol{f}_{o,s}, \boldsymbol{f}_{i\text{-}s}) + \boldsymbol{f}_{o,s}, \tag{9}$$

where $\mathcal{N}_{mhca}$ denotes the multi-head cross-attention mechanism, $\boldsymbol{f}_{o,s}$ serves as the *query* while $\boldsymbol{f}_{i\text{-}s}$ serves as the *key* and *value*, and $\boldsymbol{f}_{o\text{-}i\text{-}s} \in \mathbb{R}^{N \times 2C \times W' \times H'}$.

## 4.4 Traffic Rule Guidance

On-road object importance is also closely related with traffic rule, but it is often overlooked in previous works. To effectively leverage the traffic rule, we propose the **Traffic Rule Guidance** (**TRG**) module, which consists of two components: **lane feature extraction** and **adaptive object-lane interaction**.

**Lane feature extraction** is to make the preparation for **adaptive object-lane interaction**. In detail, a linear transformation and an activation are applied on lane information $\boldsymbol{L}$:

$$\boldsymbol{f}_l = \text{Relu}(\text{Linear}(\boldsymbol{L})), \tag{10}$$

where $\boldsymbol{L}$ are the coordinates of lane marking points, which are derived from $\boldsymbol{V}_T$ via a lane marking detector, Relu is the rectified linear unit activation, and $\boldsymbol{f}_l \in \mathbb{R}^{N \times C'}$.

**Adaptive object-lane interaction** is the core of **TRG**, and it contains two steps: *object-lane interaction* and *object-lane interaction weighting*. In the first step, lane feature $\boldsymbol{f}_l$ and $\boldsymbol{f}_{o,t}$ are fused through a multi-head cross-attention mechanism and a residual mechanism, which can be denoted as:

$$\boldsymbol{f}_{o\text{-}l}^m = \mathcal{N}_{mhca}(\boldsymbol{f}_l, \boldsymbol{f}_{o,t}) + \boldsymbol{f}_{o,t}, \tag{11}$$

where $\boldsymbol{f}_{o\text{-}l}^m \in \mathbb{R}^{N \times C'}$ denotes initial object-lane interaction feature, and $\boldsymbol{f}_l$ serves as the *query* while $\boldsymbol{f}_{o,t}$ serves as the *key* and *value*,

Factually, $\boldsymbol{f}_{o\text{-}l}^m$ has considered the traffic rule factor by modeling the relation between lane markings and on-road objects. However, the influence of lane markings on on-road object importance estimation might not be universally-effective in all scenarios, thus we propose the *object-lane interaction weighting* mechanism to realize **adaptive object-lane interaction**.

The goal of *object-lane interaction weighting* is to adaptively penalize the cases in which object-lane relation is weak (*e.g.*, static roadside cars weakly interacts with lane markings). To this end, a MLP network is applied on $\boldsymbol{f}_{o\text{-}l}^m$ to compute a score $p$, and this score is then used to compute the corresponding penalizing coefficient $p_c$, which is denoted as:

$$p = \text{Sigmoid}(\mathcal{N}_{mlp}(\boldsymbol{f}_{o\text{-}l}^m)), \tag{12}$$

$$p_c = \begin{cases} 1, & \text{if } p < 0.5 \\ \alpha, & \text{if } p \geq 0.5 \end{cases}, \tag{13}$$

where $\alpha$ is a very small value.

Based on $p_c$, the object-lane interaction feature $\boldsymbol{f}_{o\text{-}l} \in \mathbb{R}^{N \times C'}$ is obtained by weighting $\boldsymbol{f}_{o\text{-}l}^m$ in Eq. (11), which is denoted as:

$$\boldsymbol{f}_{o\text{-}l} = \boldsymbol{f}_{o\text{-}l}^m \times p_c. \tag{14}$$

We note that *object-lane interaction weighting* is the core of **adaptive object-lane interaction**, which is significant for object importance estimation (30.4% improvement on AP).

## 4.5 Object Importance Estimation

Taking $\boldsymbol{f}_{o\text{-}i\text{-}s}$ in Eq. (9) and $\boldsymbol{f}_{o\text{-}l}$ in Eq. (14) as the inputs, object importance $\boldsymbol{A} \in \mathbb{R}^N$ is estimated. This procedure is formulated as:

$$\boldsymbol{A} = \text{Softmax}(\mathcal{N}_{mlp}(\text{Linear}(\boldsymbol{f}_{o\text{-}i\text{-}s}) + \boldsymbol{f}_{o\text{-}l})), \tag{15}$$

where $\boldsymbol{A}$ signifies the importance for each object. The Linear layer transforms the dimensions of $\boldsymbol{f}_{o\text{-}i\text{-}s}$ from $N \times 2C \times W' \times H'$ to $N \times C'$ so that it can be added to $\boldsymbol{f}_{o\text{-}l}$.

# 5 Experiments

*Metrics*. For performance evaluation, two classical metrics are chosen: Average Precision (AP) and F1 Score (F1). AP is computed by calculating the area under the precision-recall curve at various thresholds, thus it is a compressive metric to indicate both the precision and recall of a model. F1 is computed by precision and recall at a fixed threshold, thus it indicates the balance between precision and recall. Both metrics follow the principle that higher values indicate better performance.

*Loss Function*. The loss function is defined as follow:

$$\mathcal{L}(\hat{A}, A) = \text{BCELoss}(\hat{A}, A) + \text{FocalLoss}(\hat{A}, A), \qquad (16)$$

where $A$ is the predicted object importance, $\hat{A}$ is the ground-truth.

## 5.1 Comparison Experiment

*Baselines*. Our model is compared with seven models. Ohn-Bar (33), Goal (8), Zhang (50), and Li (21) are representative works for on-road object importance estimation. In addition, considering that the salient object detection indicates important objects to the certain extent, three recently-proposed salient object detection models, namely MENet (23), A2S (52), and PGNet (47), are also selected as baselines.

Table 2: Quantitative comparison with baselines on TOI and Ohn-Bar (33) datasets. The 'Video' signifies the usage of RGB video sequence, 'Velocity' denotes the incorporation of vehicle velocity information, and '3D-Object' indicates the utilization of 3D object properties information.

| Method | Inputs | | | TOI | | | Ohn-Bar | | | Speed (ms/clip) |
|---|---|---|---|---|---|---|---|---|---|---|
| | Video | Velocity | 3D-Object | AP↑ | F1↑ | Acc↑ | AP↑ | F1↑ | Acc↑ | |
| Yolo | | | | 5 | 9 | 14 | 25 | 48 | 34 | 278 |
| Ohn-Bar (33) PR'2017 | ✓ | ✓ | ✓ | 19 | 28 | 74 | 39 | 14 | 62 | 100 |
| Goal (8) ICRA'2019 | ✓ | ✓ | | 50 | 49 | 90 | 52 | 26 | 65 | 155 |
| Zhang (50) ICRA'2020 | ✓ | | | 16 | 0 | 91 | 28 | 10 | 54 | 200 |
| Li (21) ICRA'2022 | ✓ | ✓ | | 23 | 0 | 91 | 41 | 0 | 64 | 202 |
| MENet (23) CVPR'2023 | ✓ | | | - | 26 | 76 | - | 12 | 62 | 122 |
| A2S (52) CVPR'2023 | ✓ | | | - | 9 | 78 | - | 18 | 61 | 120 |
| PGNet (47) CVPR'2022 | ✓ | | | - | 30 | 84 | - | 27 | 55 | 130 |
| Ours | ✓ | ✓ | | **60** | **54** | **92** | **64** | **53** | **69** | 115 |

*Quantitative Comparison*. Tab. 2 shows the comparison results on **TOI** and Ohn-Bar (33) datasets. We can observe that our model outperforms all seven baselines. On the Ohn-Bar (33) dataset, our model achieves **23.1%** and **96.3%** performance improvements on AP and F1 metrics compared with the second-best result, respectively. On the **TOI** dataset, compared with the second-best result on AP and F1 metrics, our model achieves **20.0%** (*i.e.*, (60-50)/50) and **10.2%** performance improvements, respectively. The results of MENet (23), A2S (52), and PGNet (47) are not reported on AP metric due to the lack of confidence scores in salient object detection outputs, making it impossible to calculate precision-recall curves for different thresholds. The F1 results for Zhang (50) and Li (21) are both 0 because they predict all objects as unimportant.

## 5.2 Ablation Studies

*Top-down and Bottom-up Framework*. To validate the effectiveness of our model that interactively integrates multi-fold top-down guidance mechanisms with bottom-up features, we conduct four experiments: **#1:** only the bottom-up module is enabled; **#2:** the bottom-up module is combined with **TRG** module; **#3:** the bottom-up module is combined with **DISG** module; **#4:** the bottom-up module is combined with both **TRG** module and **DISG** module. The experimental results are reported in Tab. 3.

Compared with **#1**, **#2** achieves 55% and 40% performance improvements on AP and F1, respectively. Similarly, **#3** obtains 160% and 56% performance improvements on AP and F1, respectively. When both modules are enabled (**#4**), our model exhibits the best performance. These results validate both **TRG** and **DISG** modules are effective.

Table 3: Ablation study on top-down and bottom-up framework.

| Method | BU | TRG | DISG | AP↑ | F1↑ |
|---|---|---|---|---|---|
| #1 | ✓ | | | 20 | 25 |
| #2 | ✓ | ✓ | | 31 | 35 |
| #3 | ✓ | | ✓ | 52 | 39 |
| #4 | ✓ | ✓ | ✓ | 60 | 54 |

***Driver Intention and Semantics Guidance*** (**DISG**). To verify the effectiveness of semantic context guidance and driver intention guidance, we conduct three experiments: **#1**: the model without semantic guiding feature and intention guiding mask; **#2**: the model only with semantic guiding feature; **#3**: the model only with intention guiding mask; **#4**: the model with both semantic guiding feature and intention guiding mask. The results are shown in Tab. 4. Compared to **#1**, **#2** yields 58.1% and 37.1% performance improvements on AP and F1 metrics, respectively. This enhancement is attributed to the usage of the semantic guiding feature, which enables the model to learn the

Table 4: Ablation study on DISG.

| Method | Seman. | Intent. | AP↑ | F1↑ |
|--------|--------|---------|-----|-----|
| #1 | | | 31 | 35 |
| #2 | ✓ | | 49 | 48 |
| #3 | | ✓ | 35 | 39 |
| #4 | ✓ | ✓ | **60** | **54** |

semantic relation between objects and the whole scene. Meanwhile, **#3** achieves 15.4% and 11.4% performance improvements on AP and F1 metrics, respectively. The effectiveness of our intention guiding mask accounts for this advancement. **#4** exhibits the best performance, demonstrating the effectiveness of our proposed semantic guiding feature and intention guiding mask.

***Traffic Rule Guidance*** (**TRG**). To analyze the effects of *object-lane interaction* and *object-lane interaction weighting*, we conduct three experiments: **#1**: the model without *object-lane interaction* and *object-lane interaction weighting*; **#2**: only the *object-lane interaction* is enabled; **#3**: both the *object-lane interaction* and the *object-lane interaction weighting* are enabled. The corresponding results are summarized in Tab. 5. Compared to **#1**, **#3** obtains 15.4% and 38.5% improvements on the AP and F1 metrics, respectively. These results demonstrate the significance of both *object-lane interaction* and

Table 5: Ablation study on TRG.

| Method | Interac. | Weight. | AP↑ | F1↑ |
|--------|----------|---------|-----|-----|
| #1 | | | 52 | 39 |
| #2 | ✓ | | 46 | 45 |
| #3 | ✓ | ✓ | **60** | **54** |

the *object-lane interaction weighting* mechanisms. The reason is explainable. When lane information is not utilized, the implicit traffic rule conveyed by the lane is not used. The absence of the traffic rule results in a reduced ability of the model.

It comes as a surprise that the individual usage of *object-lane interaction* (**#2**) leads to the performance decreasing on the AP metric compared to **#1**. This is due to the fact that not all on-road objects are influenced by lanes. Without *object-lane interaction weighting*, individual *object-lane interaction* generates a uniform object-lane interaction feature, which could not adaptively extend to diverse scenarios. This result potentially proves the significance of our *object-lane interaction weighting*, which enables the model to adaptively disable the object-lane interaction feature when object importance weakly rely on *object-lane interaction* (*e.g.*, static cars on the roadside).

To further analyze *object-lane interaction weighting*, we visualize its output (*i.e.*, $p_c$ in Eq. (13)). Some examples are illustrated in Fig. 3 where objects with blue masks are penalized (*i.e.*, object-lane interaction is disabled, $p_c=\alpha$) and objects with yellow masks are not penalized (*i.e.*, object-lane interaction is enabled, $p_c=1$). In Fig. 3a and Fig. 3b, the *object-lane interaction weighting* penalizes the cars

on both sides of the road. The results make sense since the static cars on roadsides are factually not interacting with lanes. In Fig. 3c and Fig. 3d, oncoming cars from the opposite direction and the car on the current lane are not penalized, since these cars are interacting with lanes. We note the yellow mask do not signal the important object. Instead, it indicates that the *object-lane interaction* is enabled.

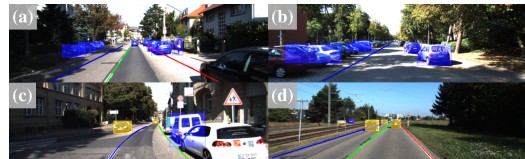

Figure 3: Visualization of *object-lane interaction weighting*.

## 6 Conclusion

On-road object importance estimation is significant for various applications in the fields of assisted driving and autonomous driving. The dilemmas of current research are two fold: **1)** the scarcity of large-scale publicly available datasets hinder the development of on-road object importance estimation, and **2)** existing methods are relatively simple to handle complex and diverse traffic scenarios. In response to the dilemmas, this paper contributes a new dataset and proposes a model with multi-fold top-down guidance. A large range of experiments demonstrate the superiority of our proposed model. The main conclusion is that building up the model that comprehensively considers multi-fold top-down guidance (*e.g.*, driver intention, semantic context, and traffic rule) and bottom-up feature (*e.g.*, size, distance, and speed) is a promising way to remarkably push forward the study of on-road object importance estimation.

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

# Appendix

This appendix provides specialized terms explanation, additional experimental results, limitations, and experimental details, which are organized as follows:

## A    Explanation of Specialized Terms

**1) Bottom-up feature**: the low-level information extracted directly from the input images or video frames using backbone networks.

**2) Top-down guidance**: the high-level information including semantic understanding, prior knowledge, specific goals, etc.

**3) Ego-car**: the car capturing video sequences that are used as the input of the model.

**4) Intention driving path**: the path from the ego-car current position to the intention destination.

**5) Intention behaviors**: the actions that the driver intends to perform based on their goals (*e.g.*, turning left, going straight, and turning right).

## B    Additional Experimental Results

### B.1    Qualitative Comparison

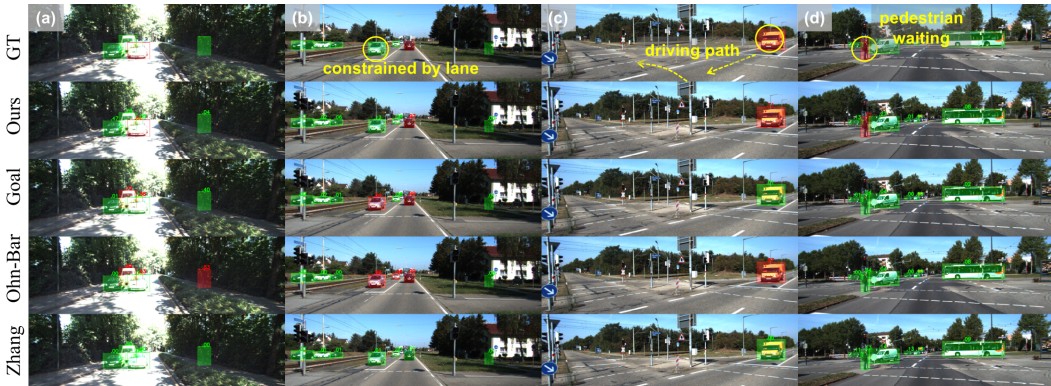

Figure 4: Qualitative comparison with baselines (*i.e.*, Goal (8), Ohn-Bar (33), and Zhang (50)). Red boxes represent important objects and green boxes denote unimportant objects.

Four scenarios (a)-(d) are illustrated in Fig. 4. In regular scenarios such as Fig. 4a, most methods function well. However, in complex scenarios, our method exhibits significant superiority. For example, in Fig. 4b, the white car on the opposite lane poses no immediate threat since it should obey the traffic rule to not drive across the solid lane marking. Baselines falsely predict it as an important object as they neglect the influence of traffic rule on object importance, while our method provides the correct estimation by considering the object-lane interaction relation in **TRG**. In Fig. 4c, the ego-car is turning left, and the vehicle on the right side is important because its driving path will risky collide with the intention driving path of ego-car. Our method successfully predicts the important object under the guidance of driver intention, while other methods fail. In Fig. 4d, a pedestrian is waiting to cross the road. Her intention walking path collides with ego-car's intention driving path, thus the pedestrian is important. Our model correctly classifies her as important thanks to the consideration of bottom-up feature and top-down guidance.

## B.2 Ablation Study on Object Feature Extraction (OFE)

To validate the rationality of **OFE**, we conduct three experiments, **#1:** only object spatial feature is used; **#2:** only object temporal feature is used; **#3:** both object features are used. The results in Tab. 6 indicate that both spatial and temporal feature of objects serve as valid foundations for accurately evaluating object importance. Removing either of them will lead to the performance decreasing, validating the reasonableness of **OFE** module. The reasons are self-evident, traffic scenarios are highly

Table 6: Ablation study on OFE.

| Method | Spat. | Temp. | AP↑ | F1↑ |
|--------|-------|-------|-----|-----|
| #1 | ✓ | | 34 | 35 |
| #2 | | ✓ | 14 | 15 |
| #3 | ✓ | ✓ | **60** | **54** |

dynamic and diverse, thus object temporal feature, which reflects motions and behaviors, is significant for object importance estimation. At the same time, object spatial feature conveys rich information such as size, distance, and orientation, thus it is also significant for object importance estimation.

## B.3 Hyperparameter Selection

We perform the experiments on hyperparameter selection, and the results are reported in Tab. 7. In the hyperparameter selection for parameters $a$ and $b$, we choose the optimal values, $a = 1, b = 1.5$, as the hyperparameters for our model. In the experiments for selecting the hyperparameter $\alpha$, it is observed that the impact of $\alpha$ is minimal across the three tested values, indicating that our method is robust.

Table 7: Ablation studies on hyperparameter selection.

| Parameter | AP↑ | F1↑ | Parameter | AP↑ | F1↑ |
|-----------|-----|-----|-----------|-----|-----|
| $a = 1, b = 2.5$ | 51 | 41 | $\alpha = 0.1$ | 57 | 45 |
| $a = 1, b = 2$ | 56 | 51 | $\alpha = 0.01$ | 55 | 52 |
| $a = 1, b = 1.5$ | 60 | 54 | $\alpha = 0.001$ | 60 | 54 |
| $a = 1, b = 1$ | 49 | 48 | | | |

## C   Limitations

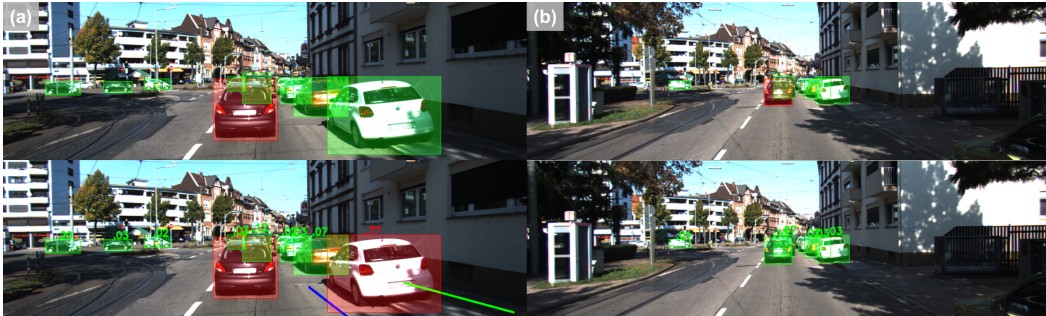

Figure 5: Failure examples. Top row is GT and bottom row is object importance estimation.

Poor lane marking detection results will limit the performance of our model. In the experiments, we find two kinds of failures. First, as shown in Fig. 5a, lane markings of current lane are falsely detected, thus the white car on the right side is supposed to locate in front of ego-car, leading to the false estimation for the white car. Second, as shown in Fig. 5b, the ego-car moves at a slow speed and lane markings are no detected, thus the model estimates the car in front of the ego-car as an unimportant parking car. Additionally, considering the effect of lane markings on on-road object importance is a small but important step towards modeling the traffic rule for this task. In the future, more traffic rule needs to be considered.

Currently, our model only considers the effect of three types of driver intentions (*i.e.*, turning left, turning right, going straight) on object importance estimation. However, real-world driving scenarios are much more complex. In future work, fine-grained intentions need to be modeled.

# D  Experimental Details

## D.1  Model Structure Details

Specific details of our model components are introduced in Tab. 8.

Table 8: **Network architecture of our model.** For Region of Interest pooling layer (ROI pooling), we list the output shape. For average pooling (Avg pooling), we list pooling scale. For Long Short-Term Memory network (LSTM), we list input and output dimension and layer number. For multi-head self-attention layer (MHSA), we list the hidden size and the head number. For multi-head cross-attention layer (MHCA), we list the hidden size and the head number. For linear layer (Linear), we list the input and output dimension. For Layer Normalization layer (LN), we list the channel dimension. For multi-layer perceptron network (MLP), we list the input channel dimension and each hidden channel dimension. Note that we use different background colors in the table to distinguish between different modules in our model, where orange represents the **OFE** module, blue represents the **DISG** module, green represents the **TRG** module, and gray represents the **OIE** module.

| Layer | Details |
|---|---|
| 1-17 | ResNet18(The last FC layer is removed) |
| 18-34 | ResNet18(The last FC layer is removed) |
| 35 | ROI pooling(10) |
| 36 | Avg pooling(16) |
| 37 | ROI pooling(10) |
| 38 | Avg pooling(16) |
| 39-40 | LSTM($512{\times}10{\times}10$, 256, 2) |
| 41 | MHSA(1024, 8) |
| 42-43 | LSTM($512{\times}10{\times}10$, 256, 2) |
| 44 | MHSA(512, 8) |
| 45 | Linear(512, 256), LN(256), ReLU |
| 46 | MHCA(1024, 8) |
| 47 | Linear($1024{\times}10{\times}10$, 256), LN(256), ReLU |
| 48 | Linear($20{\times}4$, 256), LN(256), ReLU |
| 49 | MHCA(256, 8) |
| 50 | MLP(256, {1}), Sigmoid |
| 51-52 | MLP(256, {128, 2}), Softmax |

## D.2  Implementation Details

Before the training and inference stages, we utilize CLRNet (51) model with backbone of ResNet101 and pretrained on the CULane dataset to get the $L$ in Eq. (10), and the $G$ in Eq. (5) are generated by applying DeepLabv3 (3). We use OpenCV libary to get the $M$ in Eq. (2). During the training and inference stages, we resize $V$ and $M$ in Eq. (2), and $G$ in Eq. (5) to $320{\times}320$ (*i.e.*, $W{\times}H{=}320{\times}320$). We set the $W'{=}10$, $H'{=}10$, $C{=}512$, and $C'{=}256$. The $a$ and $b$ in Eq. (6) are set as 1 and 1.5, respectively. Then, the $\beta$ in Eq. (7) is set as 2.2 since the ego-car undergoes steering when the angular velocity is around $\pm\,2.2$ based on the statistical analysis of the IMU data. In addition, the size of $m_l$, $m_s$, and $m_r$ is $10 \times 10$ (excluding the channel dimension), which is targeted to align with the size of $f_s$ (Eq. (8)). The length of video clip is set as 16. We use SDG optimizer with a weight decay of $5\mathrm{e}^{-4}$ and a momentum of 0.9. We set the batch size as 8, and use the cosine learning strategy with an initial learning rate of $1\mathrm{e}^{-4}$. Our model implementation is based on PyTorch, and experiments are conducted using an NVIDIA RTX3090 GPU.

## D.3  Object Bounding Boxes Are Assumed Known

We assume the ground truth of object bounding boxes are given. The purpose is to focus on the on-road object importance estimation while minimizing the influencing factors from the upstream object detection task. This setup is consistent with the existing on-road object importance estimation method (21). Additionally, this setup is reasonable since current object detection models are highly advanced and capable of detecting most objects on the road.

