# OpenReview forum: "On-Road Object Importance Estimation: A New Dataset and A Model with Multi-Fold Top-Down Guidance"
_NeurIPS.cc/2024/Conference — NeurIPS 2024 poster_

### Official Review · Reviewer_mvxo · 2024-07-08

**Soundness:** 3
**Presentation:** 4
**Contribution:** 3
**Rating:** 7
**Confidence:** 3

**Summary:**

1.This paper this contributes a new large-scale dataset named Traffic Object Importance (TOI) to addresses the problem of on-road object importance estimation, which utilizes video sequences captured from the driver’s perspective as the input.
2.The author also proposes a model that integrates multi-fold top-down guidance with the bottom-up feature.

**Strengths:**

1.This paper describes in great detail the specialized methodology and the structure of the models.

2.The scarcity of large-scale publicly available datasets hinder the development of on-road object importance estimation.

3. This paper considers the effect of traffic rule on object importance and successfully models this abstract concept by proposing an adaptive object-lane interaction mechanism.

**Weaknesses:**

1.In page 3 the author mentions that the traffic rule is crucial for object importance and focus on the traffic line rules , but the influence of traffic rules is varied, such as signalization. Therefore, in page 4 of table 1, the author is able to provide statistics on the scenario categories of TOI dataset and the traffic rule constraints within the dataset in experiment.

2.In page 6, the author uses three common intention behaviors in driving to reflect the driver intention (i.e., turning left, going straight, and turning right). Since the video clip length is set at 16 frames, it is important to clarify if each of the three intentions corresponds to individual frames with the 16-frame clip cut during the training and testing phases, or if multiple intentions are present within the 16 frames. The authors should further elaborate and provide the proportion of each intention in the dataset.

3.Insufficient evaluation of indicators in the experimental section. The author may add another evaluation indicator.

4.The section three can include a schematic diagram of the annotation process for the dataset.

**Questions:**

1.I consider whether 16 frames constitute interval sampling or continuous sampling, and how many types of intentional behaviors can be expressed using 16 frames in the paper.

2.The author may can add another evaluation metric for the experiment.

**Limitations:**

Yes.

---

> ### Author Rebuttal · Authors · 2024-08-07
>
> # To Reviewer mvxo
>
> Thank you very much for your positive comments and we appreciate your thoughtful feedback and suggestions.
>
> > ***Q1:**
> In page 3 the author mentions that the traffic rule is crucial for object importance and focus on the traffic line rules, but the influence of traffic rules is varied, such as signalization. Therefore, in page 4 of table 1, the author is able to provide statistics on the scenario categories of TOI dataset and the traffic rule constraints within the dataset in experiment.*
>
> **A1:**
> Thanks for this kind suggestion. We have added scenario categories and traffic rules to Table. 1 in our revised manuscript.
>
> ---
>
> > ***Q2:**
> In page 6, the author uses three common intention behaviors in driving to reflect the driver intention (i.e., turning left, going straight, and turning right). Since the video clip length is set at 16 frames, it is important to clarify if each of the three intentions corresponds to individual frames with the 16-frame clip cut during the training and testing phases, or if multiple intentions are present within the 16 frames. I consider whether 16 frames constitute interval sampling or continuous sampling, and how many types of intentional behaviors can be expressed using 16 frames in the paper. The authors should further elaborate and provide the proportion of each intention in the dataset.*
>
> **A2:**
> We apologize for any confusion. In both training and testing phases, 16 frames inputted to the model are continuous images. 16 frames occupy 1.6s in the temporal dimension, which is actually in a short time. Therefore, they share the same kind of intention. According to your suggestion, as shown in Fig. R4 of the rebuttal PDF file, we provide the proportion of each intention in the whole dataset, and we also provide the statistics of driving intentions in each scene.
>
> ---
>
> > ***Q3:**
> The author may can add another evaluation metric for the experiment.*
>
> **A3:**
> Thank you very much for this suggestion. According to your suggestion, we add another evaluation metric (i.e., accuracy), as shown in Tab. R1 of the rebuttal PDF file.
>
> ---
>
> > ***Q4:**
> The section three can include a schematic diagram of the annotation process for the dataset.*
>
> **A4:**
> Thank you very much for this suggestion. We illustrate the annotation process, as shown in Fig. R1 of the rebuttal PDF file.

---

### Official Review · Reviewer_Yrjv · 2024-07-12

**Soundness:** 3
**Presentation:** 3
**Contribution:** 3
**Rating:** 6
**Confidence:** 4

**Summary:**

This paper collects a new large-scale dataset and proposes a novel method that integrates multi-fold top-down guidance with the bottom feature to address the problem of on-road object importance estimation. Specifically, the dataset is almost three times larger than the current publicly dataset for on-road object importance. In addition, this paper considers an adaptive mechanism for object-lane interaction, effectively modeling the impact of traffic rules on object importance. Experiments on several benchmarks validate the effectiveness of the proposed method.

**Strengths:**

This paper makes several key contributions and demonstrates strengths for on-road object importance estimation

(1) This paper introduces a novel, extensive dataset, set to be released to the public, which is nearly three times the size of the current largest public dataset.

(2) The method is well-motivated and straightforward. It estimates the importance of objects on the road, integrating various top-down guidance factors with bottom-up features, marking the first of its kind.

(3) The proposed method addresses the pivotal role of traffic rules in estimating object importance, an aspect previously overlooked by existing methods. It successfully encapsulates this concept through an innovative, adaptive mechanism for object-lane interaction.

**Weaknesses:**

This paper has also two weaknesses:

(1) The paper does not provide a detailed discussion on the computational efficiency of the proposed method, which is crucial for real driving scenarios. Moreover, it is recommended to compare the model parameters and latency with other methods.

(2) Another concern lies in the practicality of the method. This method and the proposed dataset are both for single-camera scenarios, but in real autonomous driving scenarios, surrounding view is a more widely used type and a safer option. Will the proposed method also work well in the surrounding view?

**Questions:**

(1) Can the proposed method be applied to surrounding view images? I suggest that the authors should consider the application on the current perception pipeline for vision-based autonomous driving pipeline.

(2) I suggest that the authors should analyze the latency of the proposed method, which determines whether the method can be integrated into the practical driving scenarios.

**Limitations:**

The proposed method only considers the effect of three types of driver intentions on object importance estimation, which is not sufficient for complex driving scenarios. I carefully checked the paper and found no potential negative societal impact.

---

> ### Author Rebuttal · Authors · 2024-08-07
>
> # To Reviewer Yrjv
>
> Thank you very much for your positive comments and we appreciate your thoughtful feedback and suggestions.
>
> > ***Q1:**
> The paper does not provide a detailed discussion on the computational efficiency of the proposed method, which is crucial for real driving scenarios. Moreover, it is recommended to compare the model parameters and latency with other methods.*
>
> **A1:**
> According to your suggestion, we compute the latency and parameters of our method and other methods, as reported in Tab. R1 of the rebuttal PDF file. Our model asks for relatively large parameters (i.e., 173M) because multi-fold top-down guidances are involved in the model. However, 173M parameters only occupy a fraction of storage space. Therefore, the parameters will not hinder the application of our method. In comparison, the latency of a model largely affects its deployment on practical platforms. We can observe that, the latency of our method is only longer than that of Ohn-Bar. We note that Ohn-Bar is an early model using relatively simple VGG backbone, thus presenting the shortest latency.
>
> ---
>
> > ***Q2:**
> Another concern lies in the practicality of the method. This method and the proposed dataset are both for single-camera scenarios, but in real autonomous driving scenarios, surrounding view is a more widely used type and a safer option. Will the proposed method also work well in the surrounding view? Can the proposed method be applied to surrounding view images? I suggest that the authors should consider the application on the current perception pipeline for vision-based autonomous driving pipeline.*
>
> **A2:**
> Thanks for this insightful comment. Surrounding view images in autonomous driving can be broadly categorized into three types: front view, side view, and back view. Our method might not be applicable for the side view images, since DISG (Driver Intention and Semantics Guidance) module makes use of the intention guiding mask, while this mask is mainly designed for the front view. Our method should be suitable for back view images, since they are quite similar to the front view images. Thank you again for this comment, which inspire us to extend the model to be applicable for all surrounding views in the future.

---

> > ### Comment · Reviewer_Yrjv · 2024-08-13
> > **Post rebuttal**
> >
> > Thanks for the rebuttal. Most of my concerns are well-addressed therefore I tend to keep my positive rating.

---

### Official Review · Reviewer_8N7U · 2024-07-16

**Soundness:** 3
**Presentation:** 3
**Contribution:** 3
**Rating:** 6
**Confidence:** 3

**Summary:**

This paper presents a novel dataset for on-road object importance estimation. More data about which objects are important for self-driving is included and is promised to be released. Moreover, a novel method that integrates driven intention, semantic context, and traffic rule is devised to tackle the related problem. The paper is well-written.

**Strengths:**

A new dataset is introduced with rich data and labels. The presented method is novel and shown to be effective for the studied problem. Details about the dataset and the method are comprehensive and technically sound. Results are also promising.

**Weaknesses:**

Some of the concepts lack sufficient details to explain. See questions below.

**Questions:**

(1) Regarding the task, my major concern is the definition of importance. It is shown that surrounding objects that follow the traffic rules are not considered as important. Only the objects ahead of the car or have an intersection with the ego-car's direction are important. I doubt whether this is strictly appropriate. For example, if a pedestrian walking along the road, he/she will not be considered as important. However, what if this pedestrian suddenly steps into the road ahead, potential collisions would happen. Therefore, I think a nearby walking pedestrian should be considered as important or at least recognized into a third category like "needs care". I wonder how the authors solve this problem in the dataset.
(2) Regarding the driver's intention, it is indeed difficult to define appropriately. The authors have mentioned this in the paper, but the strategy introduced to accommodate this is still not clear to me. The authors mentioned learning the intention values based on driving behaviors, but how do we know the driving behaviors? Are these behaviors (e.g. turning left) already provided in the dataset?
(3) More visualization about the labels and method comparisons are better to be presented for more clarity.

**Limitations:**

The authors have mentioned limitations.

---

> ### Author Rebuttal · Authors · 2024-08-07
>
> # To Reviewer 8N7U
>
> Thank you very much for your positive comments and we appreciate your thoughtful feedback and suggestions.
>
> > ***Q1:**
> Regarding the task, my major concern is the definition of importance. It is shown that surrounding objects that follow the traffic rules are not considered as important. Only the objects ahead of the car or have an intersection with the ego-car's direction are important. I doubt whether this is strictly appropriate. For example, if a pedestrian walking along the road, he/she will not be considered as important. However, what if this pedestrian suddenly steps into the road ahead, potential collisions would happen. Therefore, I think a nearby walking pedestrian should be considered as important or at least recognized into a third category like "needs care". I wonder how the authors solve this problem in the dataset.*
>
> **A1:**
> Our core idea to define the object importance is whether an object affects the safe driving. With this core idea in the mind, we summarize four types of importance evaluation guidelines, including object attribute, driver intention, traffic semantic context, and traffic rule, as detailed in Line50-63 in the original version. Considering the certain individual guideline might not be strictly applicable for some traffic scenarios, we introduce the double-checking and triple-discussing annotation mechanisms to comprehensively make use of four mutually-dependent guidelines to handle disputed and complex scenarios. Your above mentioned scenario is complex. In this scenario, object attribute (e.g., the distance and object category) is the dominant guideline. The closer an object is to the ego-car, the more important it is. In addition, pedestrian is a special object category, the behavior of a pedestrian is more random than a vehicle, and traffic rule imposes fewer constraints on a pedestrian than that on a vehicle. Moreover, the pedestrian is a crucial category in traffic scenes. Under the similar condition, the pedestrian is more important than other categories. Therefore, a nearby walking pedestrian should be annotated as important.
>
> ---
>
> > ***Q2:**
> Regarding the driver's intention, it is indeed difficult to define appropriately. The authors have mentioned this in the paper, but the strategy introduced to accommodate this is still not clear to me. The authors mentioned learning the intention values based on driving behaviors, but how do we know the driving behaviors? Are these behaviors (e.g. turning left) already provided in the dataset?*
>
> **A2:**
> Actually, the annotations of driving behaviors are not provided. However, the KITTI dataset contains detailed IMU/GPS data, and we can determine whether the vehicle is turning left, turning right, or going straight based on the vehicle's lateral angular velocity derived from the IMU data.

---

> > ### Comment · Reviewer_8N7U · 2024-08-09
> >
> > In fact, I think I do not get too much new information from the authors' response to my question 1, though this would not affect my original rating. Overall this is an interesting paper. I only suggest that the authors present more discussions on the annotation process, not only on how annotations are obtained but also on the strengths and weaknesses of the applied annotation policy. Regarding the answer to my question 2, I can get the idea and I also suggest adding more related discussions in the paper.

---

### Official Review · Reviewer_xrYx · 2024-07-22

**Soundness:** 3
**Presentation:** 3
**Contribution:** 2
**Rating:** 6
**Confidence:** 3

**Summary:**

This work addresses the issue of estimating the importance of on-road objects using video sequences from a driver’s perspective, a critical task for enhancing driving safety. The authors introduce the Traffic Object Importance (TOI) dataset, which is significantly larger and more diverse than existing datasets, and propose a novel model that integrates multi-fold top-down guidance factors—driver intention, semantic context, and traffic rules—with bottom-up features for more accurate importance estimation. Experimental results demonstrate that the proposed model significantly outperforms state-of-the-art methods in on-road object importance estimation.

**Strengths:**

1. The introduction of the Traffic Object Importance (TOI) dataset, which is significantly larger and more diverse than existing datasets, provides a robust foundation for training and evaluating models in on-road object importance estimation, thereby addressing a major limitation in the field.

2.  The proposed model effectively integrates multi-fold top-down guidance factors—driver intention, semantic context, and traffic rules—with bottom-up features, which showed good performance for the TOI task.

**Weaknesses:**

1. Lack of description of the annotation details.
How many annotators are involved in the annotation procedure? It would be good if the authors can provide some annotation procedure samples regarding the double-checking annotation mechanism and the triple-discussing annotation mechanism.

2. It seems this annotation will be varied according to different traffic rules. Since KITTI is collected in Germany, the annotators should be familiar to germany traffic rules. However the authors did not mention this information in their submission, thereby the label quality is doubtful.

3. The authors are encouraged to build up the first benchmark based on the proposed dataset by using various existing object detection methods, e.g., Yolo, with the proposed head or simpler head. It is interesting to see how the existing object detectors work on this new task.

4. More statistics of the dataset are encouraged to be given, e.g., the number of important object of different categories, etc.

**Questions:**

1. How many annotators were involved in the annotation procedure for the dataset? Can the authors provide detailed examples of their double-checking and triple-discussing annotation mechanisms?

2. Were the annotators familiar with German traffic rules, given that the dataset was collected in Germany (KITTI dataset)? How was the expertise of the annotators in relation to German traffic laws ensured and validated?

3. Have the authors considered building the first benchmark using their dataset with existing object detection methods, such as YOLO? What were the performance outcomes of these existing methods when applied to the new task?

4. Can the authors provide more detailed statistics about the dataset, such as the number of important objects in different categories? How do these statistics compare to other datasets in the same domain?

**Limitations:**

yes the authors mentioned it in the appendix

---

> ### Author Rebuttal · Authors · 2024-08-07
>
> # To Reviewer xrYx
>
> Thank you very much for your positive comments on our proposed dataset addressing a major limitation in the field and our model showing the good performance.
>
> > ***Q1:**
> Lack of description of the annotation details. How many annotators are involved in the annotation procedure?*
>
> **A1:**
> The details of annotating a sequence is shown in Fig. R2 of the rebuttal PDF file. We also illustrate the annotation process in Fig. R1 of the rebuttal PDF file, from which we can observe the annotation process involves three types of annotators, namely the first, second, and third annotator. Six experienced drivers are recruited as volunteers. It is worth noting that every volunteer is able to act as the role of the first, second, or third annotator.
>
> ---
>
> > ***Q2:**
> It would be good if the authors can provide some annotation procedure samples regarding the double-checking annotation mechanism and the triple-discussing annotation mechanism.*
>
> **A2:**
> To illustrate the double-checking and triple-discussing annotation mechanisms, we provide a schematic showing our annotation process in Fig. R1 of the rebuttal PDF file. To better explain double-checking and triple-discussing mechanisms, an example is provided, please see the image (with a van and a cyclist inside) in the box "discuss together" in Fig. R1. The first annotator annotates both van and cyclist as important objects, considering both objects are along the driving path. When the second annotator is checking the annotation, claiming that the annotation for the cyclist is reasonable but the annotation for the van is disputed, since van is far from the driver (though it is along the driving path) and the driver will only pay attention to the cyclist. At this time, the first and second annotators discuss together to convince each other. If they can not reach an agreement, the triple-discussing mechanism is activated, and the third annotator join the discussion. The third annotator holds the idea that the van is also a potential object that might affect the driving safety, thus the final annotation is that both van and cyclist are important objects.
>
> ---
>
> > ***Q3:**
> It seems this annotation will be varied according to different traffic rules. Since KITTI is collected in Germany, the annotators should be familiar to germany traffic rules. However the authors did not mention this information in their submission, thereby the label quality is doubtful.*
>
> **A3:**
> During our annotation process, we focus primarily on universal traffic rules (e.g., vehicles should not drive across solid lanes). Therefore, the traffic rule imposes little effect on annotations. Factually, we find the main effect comes from that many objects simultaneously exist in an image. Thus, to guarantee the reliability of annotations, the first annotator only annotates one object at each time of observing the whole sequence. The annotation for a sequence is finished until all objects are annotated, as illustrated in Fig. R2 of the rebuttal PDF file. In addition, in existing datasets, although multiple annotators perform the annotations, it is not mentioned that the initial annotations are checked by other annotators. Differently, the double-checking and triple-discussing annotation mechanisms are introduced to further guarantee the quality of our annotations. We hope our explanation alleviates your concern.
>
> ---
>
> > ***Q4:**
> The authors are encouraged to build up the first benchmark based on the proposed dataset by using various existing object detection methods, e.g., Yolo, with the proposed head or simpler head. It is interesting to see how the existing object detectors work on this new task.*
>
> **A4:**
> We have conducted the experiment that uses YOLO to build up a benchmark. To make YOLO suitable for our task, we adjust the output dimensions of the final fully connected layer in YOLO detection head to 2 to indicate important and unimportant. The experimental results are reported in Tab. R1 of the rebuttal PDF file.
>
> ---
>
> > ***Q5:**
> Can the authors provide more detailed statistics about the dataset, such as the number of important objects in different categories? How do these statistics compare to other datasets in the same domain?*
>
> **A5:**
> We present more statistics about the dataset in Fig. R4 of the rebuttal PDF file, including the statistics on different driving intentions and the number of important objects in different object categories. Unfortunately, due to the missing information in other datasets, we could not provide the corresponding information of other datasets.

---

> > ### Comment · Reviewer_xrYx · 2024-08-09
> > **Response to the authors**
> >
> > Thank you for your response. My concerns are mostly solved. I would lile to improve my score to 6.

---

### Author Rebuttal · Authors · 2024-08-07

# General Response
We thank reviewers for their valuable feedback. We are encouraged by the reviewers’ positive comments on our work. Specifically, they find our model novel (8N7U) and effective (xrYx, 8N7U), our idea well-motivated (Yrjv), our proposed dataset sound (8N7U), our paper detailed (8N7U, mvxo) and well-written (8N7U). Reviewer xrYx remarks that *"provides a robust foundation for training and evaluating models in on-road object importance estimation, thereby addressing a major limitation in the field"*.
Reviewer Yrjv comments that *“The proposed method addresses the pivotal role of traffic rules in estimating object importance”*.

We will provide detailed responses to reviewers' questions and feedback, hoping to address any confusion and concern.

---

### Decision · Program_Chairs · 2024-09-25

**Decision:**

Accept (poster)

**Comment:**

## Summary

The Traffic Object Importance (TOI) dataset is introduced to estimate the importance of on-road objects from a driver's perspective, a crucial task for enhancing driving safety. The dataset is larger and more diverse than existing ones, and a novel model is proposed that integrates driver intention, semantic context, and traffic rules with bottom-up features for more accurate estimation. Experimental results show that the proposed model significantly outperforms state-of-the-art methods in on-road object importance estimation. The paper also introduces an adaptive mechanism for object-lane interaction.

## Strengths
-  Introduces a larger and diverse dataset for on-road object importance estimation.
- Integrates top-down guidance factors (driver intention, semantic context, traffic rules) with bottom-up features.
- Shows promising performance for the TOI task.
- Addresses the pivotal role of traffic rules in estimating object importance.
- Proposes an adaptive object-lane interaction mechanism.

## Weaknesses

- Lack of detailed annotation details, including the number of annotators involved and the annotation procedure samples.
- Annotation may vary according to different traffic rules, which could affect label quality.
- Definition of importance is unclear, with only objects ahead of the car or with an intersection with the ego-car's direction considered important.
- Driver's intention definition is difficult to define, with unclear strategies for learning intention values based on driving behaviors.
- More statistics of the dataset are needed, such as the number of important objects of different categories.

## Conclusion

The paper proposed (i) a dataset that is larger and more diverse than existing ones, and (ii)  a novel model  that integrates driver intention, semantic context, and traffic rules with bottom-up features for more accurate estimation. Therefore, the recommendation is the acceptance of the paper but it should include a benchmark with the current object detectors.